# New Advancements in AdS/CFT in Lower Dimensions

**Yolanda Lozano** [1,2,*,†] **and Anayeli Ramirez** [1,2,†]

1   Department of Physics, University of Oviedo, Avda. Federico Garcia Lorca s/n, 33007 Oviedo, Spain;
    ramirezanayeli.uo@uniovi.es
2   Instituto Universitario de Ciencias y Tecnologías Espaciales de Asturias (ICTEA),
    Calle de la Independencia 13, 33004 Oviedo, Spain.
*   Correspondence: ylozano@uniovi.es
†   These authors contributed equally to this work.

**Abstract:** We review recent developments in the study of the AdS/CFT correspondence in lower dimensions. We start by summarising the classification of $AdS_3 \times S^2$ solutions in massive type IIA supergravity with (0, 4) supersymmetries and the construction of their 2D dual quiver CFTs. These theories are the seed for further developments that we review next. First, we construct a new class of $AdS_3$ solutions in M-theory that describe M-strings in M5-brane intersections. Second, we generate a new class of $AdS_2 \times S^3$ solutions in massive IIA with four supercharges that we interpret as describing backreacted baryon vertices within the 5D $\mathcal{N} = 1$ QFT living in D4-D8 branes. Third, we construct two classes of $AdS_2$ solutions in Type IIB. The first are dual to discrete light-cone quantised quantum mechanics living in null cylinders. The second class is interpreted as dual to backreacted baryon vertices within 4D $\mathcal{N} = 2$ QFT living in D3-D7 branes. Explicit dual quiver field theories are given for all classes of solutions. These are used to compute the central charges of the CFTs that are shown to agree with the holographic expressions.

**Keywords:** string theory; AdS/CFT correspondence; supergravity

## 1. Introduction

Lower dimensional CFTs [1] play a prominent role in the microscopic description of black holes and black strings. Since these exhibit, in the extremal case, $AdS_2$ and $AdS_3$ geometries close to their horizons, a deeper understanding of the AdS/CFT correspondence in lower dimensions is of key importance to their study.

The construction of $AdS_2$ and $AdS_3$ geometries and the identification of their dual superconformal field theories has been the focus of many interesting works. In general, the possible geometrical structures, supersymmetries preserved and topologies of the solutions increase as one increases the dimensions of the internal space, giving rise to a plethora of possible solutions (see for instance [1–45]). However, even if many classes of solutions with different amounts of supersymmetries have been constructed, this has only been paralleled with a detailed understanding of their dual CFTs for D1-D5 and D1-D5-D5' systems, and orbifolds thereof [16,46–51].

In this work, we review recent progress on the construction of $AdS_3/CFT_2$ and $AdS_2/CFT_1$ pairs with four supersymmetries where both sides of the correspondence are reasonably well-understood. These provide controlled settings where the AdS/CFT correspondence can be explicitly checked and where the black hole microstate counting programme can be carried out in detail.

Important progress in our understanding of the $AdS_3/CFT_2$ correspondence was provided by the recent constructions in [26]. These are solutions to massive Type IIA supergravity with $\mathcal{N} = (0, 4)$ supersymmetries (and SU(2) R-symmetry) that are realised as fibrations of $AdS_3 \times S^2 \times M_4$ on an interval, with $M_4$ either a $CY_2$ or a 4d Kähler manifold. These solutions are dual to interesting classes of 2D CFTs admitting a quiver description in the UV that can be used to compute their degrees of freedom [52–54].

Besides providing explicit $AdS_3/CFT_2$ pairs, the constructions in [26,52–54] have been the seed of many other interesting developments. In a nutshell, new classes of $AdS_3$ solutions in M-theory with the same number of supersymmetries have been constructed in [34], which provide the holographic duals of the configurations of M-strings described in [55,56]. From the latter, new classes of solutions in massless Type IIA have been generated [36], which allow for an explicit defect interpretation as surface quivers embedded in a 6D CFT. Perhaps more interestingly, new examples of explicit $AdS_2/CFT_1$ duals have been derived from these $AdS_3/CFT_2$ pairs in both Type II supergravities [38,40,43].

The construction of new $AdS_2/CFT_1$ pairs is of special relevance. $AdS_2$ geometries arise as near horizon geometries of extremal black holes and therefore play a very important in their microscopical studies. However, it is well-known that the precise realisation of an $AdS_2/CFT_1$ correspondence presents important technical and conceptual problems [57–60] that mainly originate from the fact that $AdS_2$ possesses two disjoint boundaries [61]. As a result, this correspondence is in need of a much deeper understanding.

A successful approach taken in [38,40,43] was to exploit its connection with the much better understood $AdS_3/CFT_2$ correspondence. At the geometrical level, $AdS_3$ and $AdS_2$ spaces are related by T-duality. This allows one to construct explicit $AdS_2/CFT_1$ pairs in which the $CFT_1$ arises as a discrete light-cone compactification of the 2D CFT dual to the $AdS_3$ solution, thus providing an explicit realisation of the constructions in [62–65]. Moreover, if two spheres are present in the internal space of an $AdS_3$ solution, such solutions are amenable to double analytical continuation techniques, which turn $AdS_3 \times S^2$ spaces into $AdS_2 \times S^3$ geometries. This latter class of solutions can then be dual to more general superconformal quantum mechanics (SCQM) than those arising from discrete light-cone compactification.

The aim of this review article is to summarise the main features of these recent developments. The paper is organised as follows. We start in Section 2 by reviewing the $AdS_3/CFT_2$ pairs constructed in [26,52–54], seeds of the forthcoming constructions. In Section 3, we review their uplift to M-theory, following [34], and briefly address their interpretation as duals to the M-strings in [55,56]. In Section 4, we turn our attention to the construction of new $AdS_2/CFT_1$ pairs in massive Type IIA, following [40]. We describe in some detail the associated dual SCQM, which allows one to interpret the solutions as backreacted D4-D0 baryon vertices in the 5D CFT living in D4′-D8 brane intersections. In Section 5, we discuss two more $AdS_2/CFT_1$ pairs, in this case in Type IIB, following [38,43]. A first class of solutions is constructed from the seed $AdS_3$ solutions reviewed in Section 2 using T-duality. These solutions are dual to explicit 1D CFTs that arise as discrete light-cone compactifications of the 2D CFTs reviewed in Section 2. The second class of solutions is constructed by T-dualising the $AdS_2$ class reviewed in Section 4. These solutions allow for an interesting interpretation as backreacted D5-D1 baryon vertices in the 4D $\mathcal{N} = 2$ QFT living in D3-D7 brane intersections, that we briefly discuss. Finally, in Section 6, we discuss open lines for further investigation.

## 2. $AdS_3/CFT_2$ in Massive IIA

In [26], a thorough classification of $AdS_3 \times S^2$ solutions to massive IIA supergravity consistent with a nontrivial Romans mass, with small (0, 4) supersymmetry and SU(2)-structure, was obtained. The solutions are warped products of the form $AdS_3 \times S^2 \times M_4 \times I$, where $M_4$ is either a $CY_2$ or a 4D Kähler manifold. In this review, we restrict ourselves to the particular case when $M_4 = CY_2$, and there is no dependence on the coordinates of the $CY_2$. These solutions provide the "seed" from which all supergravity backgrounds summarised in this work are derived. Furthermore, we review the proposal in [52–54] for their two-dimensional dual CFTs.

The Neveu–Schwarz (NS) sector of this class of solutions is as follows:

$$ds^2 = \frac{u}{\sqrt{h_4 h_8}}\left(ds^2_{\text{AdS}_3} + \frac{h_8 h_4}{4 h_8 h_4 + u'^2}ds^2_{\text{S}^2}\right) + \sqrt{\frac{h_4}{h_8}}ds^2_{\text{CY}_2} + \frac{\sqrt{h_4 h_8}}{u}d\rho^2,$$

$$e^{-\Phi} = \frac{h_8^{\frac{3}{4}}}{2 h_4^{\frac{1}{4}}\sqrt{u}}\sqrt{4 h_8 h_4 + u'^2}, \quad H_3 = \frac{1}{2}d\left(-\rho + \frac{u u'}{4 h_4 h_8 + u'^2}\right) \wedge \text{vol}_{\text{S}^2}, \tag{1}$$

where $\Phi$ is the dilaton, $H_3 = dB_2$ is the NS-NS three-form and the metric is written in string frame. The warping functions $h_4$, $h_8$ and $u$ are functions of $\rho$, which parameterises an interval. We denote $' = \partial_\rho$. The RR fluxes are

$$F_0 = h_8', \quad F_2 = -\frac{1}{2}\left(h_8 - \frac{h_8' u' u}{4 h_8 h_4 + u'^2}\right)\text{vol}_{\text{S}^2},$$

$$F_4 = -\left(d\left(\frac{u u'}{2 h_4}\right) + 2 h_8 d\rho\right) \wedge \text{vol}_{\text{AdS}_3} - h_4' \text{vol}_{\text{CY}_2}, \tag{2}$$

with the higher fluxes related to them as $F_p = (-1)^{[p/2]} \star_{10} F_{10-p}$. The background in (1) and (2) is a solution of the equations of motion if the functions $h_4, h_8$ and $u$ satisfy

$$h_4''(\rho) = 0, \quad h_8''(\rho) = 0, \quad u''(\rho) = 0, \tag{3}$$

away from localised sources, which makes them linear functions of $\rho$. The first two equations are Bianchi identities, whereas $u'' = 0$ is a BPS equation.

The Page fluxes, defined as $\hat{F} = e^{-B_2} \wedge F$, are given by

$$\hat{F}_0 = h_8', \qquad \hat{F}_2 = -\frac{1}{2}\left(h_8 - h_8'(\rho - 2\pi k)\right)\text{vol}_{\text{S}^2},$$

$$\hat{F}_4 = -\left(\partial_\rho\left(\frac{u u'}{2 h_4}\right) + 2 h_8\right)d\rho \wedge \text{vol}_{\text{AdS}_3} - h_4' \text{vol}_{\text{CY}_2}. \tag{4}$$

Here, we included large gauge transformations of $B_2$ of parameter $k$, $B_2 \to B_2 + \pi k \text{vol}_{\text{S}^2}$, for $k = 0, 1, \ldots, P$. The transformations are performed every time a $\rho$-interval $\rho \in [2\pi k, 2\pi(k+1)]$ is crossed. They ensure that $B_2$ satisfies the condition coming from String Theory:

$$\frac{1}{4\pi^2}\Big|\int_{\text{S}^2} B_2\Big| \in [0,1]. \tag{5}$$

The most general solution to (3) is that $h_4$ and $h_8$ are piecewise linear functions. This allows for source branes in the geometry. In turn, $u$ needs to be continuous for the preservation of supersymmetry. Here, we restrict it to the simpler case in which $u = $ constant [2]. In [52–54], piecewise linear solutions with the $\rho$ direction bounded between 0 and $2\pi(P+1)$, where $h_4$ and $h_8$ vanish, were proposed. These functions read

$$h_4(\rho) = \begin{cases} \frac{\beta_0}{2\pi}\rho & 0 \le \rho \le 2\pi, \\ \alpha_k + \frac{\beta_k}{2\pi}(\rho - 2\pi k) & 2\pi k \le \rho \le 2\pi(k+1), \quad k = 1,\ldots,P-1 \\ \alpha_P - \frac{\alpha_P}{2\pi}(\rho - 2\pi P) & 2\pi P \le \rho \le 2\pi(P+1), \end{cases} \tag{6}$$

$$h_8(\rho) = \begin{cases} \frac{\nu_0}{2\pi}\rho & 0 \le \rho \le 2\pi, \\ \mu_k + \frac{\nu_k}{2\pi}(\rho - 2\pi k) & 2\pi k \le \rho \le 2\pi(k+1), \quad k = 1,\ldots,P-1 \\ \mu_P - \frac{\mu_P}{2\pi}(\rho - 2\pi P) & 2\pi P \le \rho \le 2\pi(P+1). \end{cases} \tag{7}$$

The space begins at $\rho = 0$ in a smooth fashion. In turn, at $\rho = 2\pi(P+1)$, the behaviour of the metric and dilaton is that of a superposition of D2-branes wrapped on AdS$_3$ and smeared on the CY$_2 \times$ S$^2$, and of D6-branes wrapped on AdS$_3 \times$ CY$_2$ [3].

The quantities $(\alpha_k, \beta_k, \mu_k, \nu_k)$ are integration constants and must satisfy

$$\alpha_k = \sum_{j=0}^{k-1} \beta_j, \quad \mu_k = \sum_{j=0}^{k-1} \nu_j. \tag{8}$$

in order for the metric and dilaton to be continuous. For the solutions defined by (6) and (7), the quantised charges associated with the Page fluxes given by (4) in the different $2\pi k \leq \rho \leq 2\pi(k+1)$ intervals are

$$Q_{D2}^{(k)} = \alpha_k = \sum_{j=0}^{k-1} \beta_j, \qquad Q_{D6}^{(k)} = \mu_k = \sum_{j=0}^{k-1} \nu_j$$

$$Q_{D4}^{(k)} = \beta_k, \qquad Q_{D8}^{(k)} = \nu_k, \qquad Q_{NS5}^{(k)} = 1. \tag{9}$$

The field theory duals to this class of solutions were studied in [52–54] [4]. We summarise them in the following subsection.

### 2.1. Two-Dimensional Dual CFTs

The backgrounds defined by Equations (1)–(3) are associated with the brane intersections depicted in Table 1. In these brane set-ups the D2- and D6-branes play the role of colour branes, while the D4- and D8-branes play the role of flavour branes. This is supported by the analysis of the Bianchi identities, which yields

$$dF_0 = \sum_{k=1}^{P} \left( \frac{\nu_{k-1} - \nu_k}{2\pi} \right) \delta(\rho - 2\pi k) d\rho$$

$$d\hat{F}_4 = \sum_{k=1}^{P} \left( \frac{\beta_{k-1} - \beta_k}{2\pi} \right) \delta(\rho - 2\pi k) d\rho \wedge \text{vol}_{CY_2}, \tag{10}$$

indicating that, at the point $\rho = 2\pi k$, there is the possibility of having localised D8- and D4-branes. Indeed, explicit D8- and D4-branes need to be present at $\rho = 2\pi k$ if the slopes of $h_8$ and $h_4$ are different at both sides. Their numbers are given by

$$\Delta Q_{D8}^{(k)} = \nu_{k-1} - \nu_k, \qquad \Delta Q_{D4}^{(k)} = \beta_{k-1} - \beta_k. \tag{11}$$

The associated Hanany–Witten brane set-up is then the one depicted in Figure 1.

**Table 1.** BPS brane intersection underlying the geometry given by (1)–(3). $(x^0, x^1)$ are the directions where the 2D CFT lives. The directions $(x^2, \ldots, x^5)$ span the CY$_2$, on which the D6- and the D8-branes are wrapped. The coordinate $x^6$ is the direction associated with $\rho$. Finally, $(x^7, x^8, x^9)$ are the transverse directions realising the SO(3) R-symmetry.

|       | 0 | 1 | 2 | 3 | 4 | 5 | 6 | 7 | 8 | 9 |
|-------|---|---|---|---|---|---|---|---|---|---|
| D2    | x | x |   |   |   |   | x |   |   |   |
| D4    | x | x |   |   |   |   |   | x | x | x |
| D6    | x | x | x | x | x | x | x |   |   |   |
| D8    | x | x | x | x | x | x |   | x | x | x |
| NS5   | x | x | x | x | x | x |   |   |   |   |

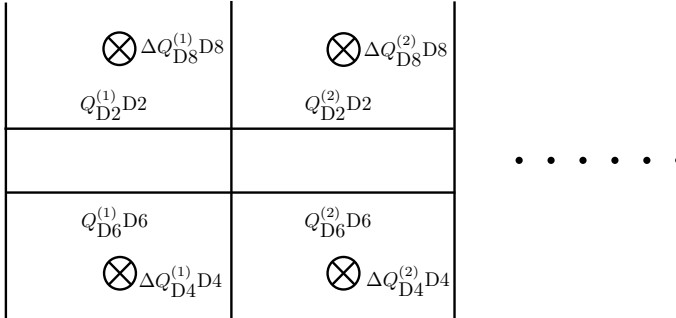

**Figure 1.** Generic Hanany–Witten brane set-up associated with the holographic background defined by the functions in (6) and (7). The vertical lines are NS5-branes, the horizontal lines represent D2- and D6-branes, and the crosses indicate D4- and D8-branes.

It was shown in [52–54] that these brane set-ups define 2D CFTs with $\mathcal{N} = (0, 4)$ SUSY. These CFTs describe the strongly coupled IR fixed points of the two-dimensional quantum field theories living in them. These field theories are encoded in the quivers depicted in Figure 2. Since the extension of the D2 and D6 branes is finite in the $\rho$ direction, the field theory that lives in their intersection is effectively two dimensional at low energies. The quivers become non-anomalous when adequate flavour groups are attached at each node, coming from D4- and D8-branes. Their dynamics is described in terms of $(0, 4)$ vector multiplets and hypermultiplets, coming from the open strings that connect the different types of branes. This analysis was presented in [69]. It differs slightly from the one originally considered in [52–54], which also led to anomaly free quivers with the same central charge to leading order but was not based directly on open string quantisation. We next summarise the main ingredients of the quivers based on open string quantisation, following [69]:

- To each gauge node corresponds to a $(0, 4)$ vector multiplet, represented by a circle, plus a $(0, 4)$ hypermultiplet in the adjoint representation of the gauge group, represented by a grey line starting and ending on the same gauge group. In terms of $(0, 2)$ multiplets, the first consists of a vector multiplet and a Fermi multiplet in the adjoint, and the second consists of two chiral multiplets forming a $(0, 4)$ hypermultiplet.
- Between each pair of horizontal nodes there are two $(0, 2)$ Fermi multiplets, forming a $(0, 4)$ Fermi multiplet, and two $(0, 2)$ chiral multiplets, forming a $(0, 4)$ twisted hypermultiplet, each in the bifundamental representation of the gauge groups. The $(0, 4)$ Fermi multiplet and the $(0, 4)$ twisted hypermultiplet combine into a $(4, 4)$ twisted hypermultiplet. They are represented by horizontal black solid lines.
- Between each pair of vertical nodes, there are two $(0, 2)$ chiral multiplets forming a $(0, 4)$ hypermultiplet in the bifundamental representation of the gauge groups. They are represented by grey lines.
- Between each gauge node and any successive or preceding node, there is one $(0, 2)$ Fermi multiplet in the bifundamental representation. It is represented by dashed lines.
- Between each gauge node and its adjacent global symmetry node, there is one $(0, 2)$ Fermi multiplet in the fundamental representation of the gauge group. It is again represented by dashed lines.
- Between each gauge node and its opposing global symmetry node, there are two $(0, 2)$ Fermi multiplets, forming a $(0, 4)$ Fermi multiplet, and two $(0, 2)$ chiral multiplets, forming a $(0, 4)$ twisted hypermultiplet, each in the fundamental representation of the gauge groups. The $(0, 4)$ Fermi multiplet and the $(0, 4)$ twisted hypermultiplet combine into a $(4, 4)$ twisted hypermultiplet. They are represented by curvy black solid lines.

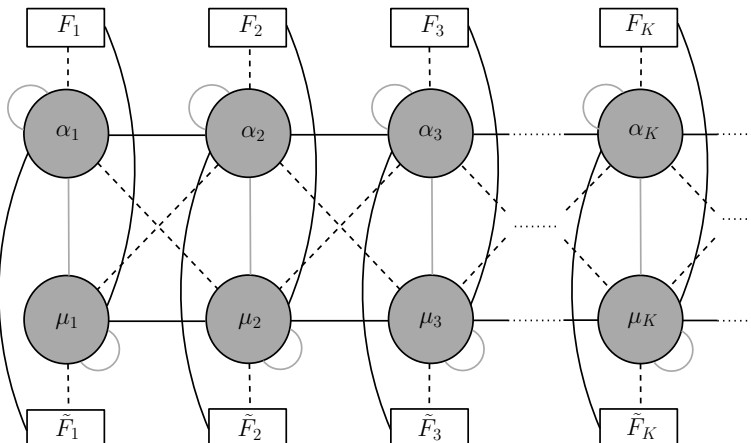

**Figure 2.** A generic quiver field theory in which the IR is dual to the holographic background defined by the functions in (6) and (7).

The previous fields contribute to the anomaly of a generic SU($N_i$) gauge group as follows:

- A (0, 2) vector multiplet contributes a factor of $-N_i$.
- A (0, 2) chiral multiplet in the adjoint representation contributes with a factor of $N_i$.
- A (0, 2) chiral multiplet in the bifundamental representation contributes with a factor of $\frac{1}{2}$.
- A (0, 2) Fermi multiplet in the adjoint representation contributes with a factor of $-N_i$.
- A (0, 2) Fermi multiplet in the fundamental of bifundamental representation contributes with a factor of $-\frac{1}{2}$.

Putting these together, we have that, for generic SU($\alpha_k$) and SU($\mu_k$) colour groups, the gauge anomaly cancellation conditions are, respectively,

$$F_k = \nu_{k-1} - \nu_k, \quad \tilde{F}_k = \beta_{k-1} - \beta_k, \tag{12}$$

for the $F_k$ and $\tilde{F}_k$ flavour groups. This is precisely the number of D8 and D4 flavour (source) branes in each interval, as shown by Equation (11).

In turn, as shown in [70], the right-handed central charge of the IR SCFT is calculated by associating it with the U(1)$_R$ current two-point function;

$$c_{cft} = 3 \operatorname{Tr}[\gamma_3 Q_R^2], \tag{13}$$

where $Q_R$ is the charge with respect to the U(1)$_R \subset$ SU(2)$_R$ and the trace is over all Weyl fermions in the theory. The two left-handed fermions inside the (0, 4) vector multiplet have R-charge equal to 1. For hypermultiplets, we have that, for both right-handed fermions inside a (0, 4) hypermultiplet, the R-charge is $-1$, while those in a (0, 4) twisted hypermultiplet have zero R-charge. Finally, the left-handed fermion inside the (0, 2) Fermi multiplet also has a vanishing R-charge. Putting this together, we find that

$$c_{cft} = 6(n_{hyp} - n_{vec}), \tag{14}$$

where $n_{hyp}$ is the number of (untwisted) (0, 4) hypermultiplets and $n_{vec}$ is the number of (0, 4) vector multiplets.

In [52–54], a number of dual holographic pairs were presented that provided stringent support for the validity of the proposed duality. In each example, the field theory central charge given by expression (14) was shown to coincide (for long quivers with large ranks,

when the background is a trustable dual description of the CFT) with the holographic central charge. This was computed from the Brown–Henneaux formula, giving

$$c_{hol} = \frac{3\pi}{2G_N} \text{Vol}_{CY_2} \int_0^{2\pi(P+1)} h_4 h_8 d\rho = \frac{3}{\pi} \int_0^{2\pi(P+1)} h_4 h_8 d\rho. \tag{15}$$

Here, we used that $G_N = 8\pi^6$, with $g_s = \alpha' = 1$, and that $\text{Vol}_{CY_2} = 16\pi^4$. For the functions $h_4$ and $h_8$ displayed above, this gives

$$c_{hol} = \sum_{k=1}^{P} \left( 6\alpha_k \mu_k + 3(\alpha_k \nu_k + \beta_k \mu_k) + 2\beta_k \nu_k \right), \tag{16}$$

which can be shown to agree in the holographic limit with the expression coming from (14).

### 3. AdS$_3$ Solutions in M-Theory

In this section, we consider the uplift to eleven dimensions of the solutions discussed in the previous section, following [34]. In order to carry out this uplift, we need to restrict it to vanishing Romans' mass, $F_0 = 0$, which imposes $h_8' = 0$, and thus both the absence of D8-branes and the presence of a constant number of D6-branes between all pairs of NS5-branes. In the uplift to eleven dimensions, this number becomes a modding parameter of the geometry, associated with the KK-monopole charge.

The M-theory solutions are of the form AdS$_3\times$S$^3/$**Z**$_k\times$CY$_2$ fibred over an interval. They read as follows:

$$
\begin{aligned}
ds_{11}^2 =&\Delta\left( \frac{u}{\sqrt{h_4 h_8}} ds_{\text{AdS}_3}^2 + \sqrt{\frac{h_4}{h_8}} ds_{CY_2}^2 + \frac{\sqrt{h_4 h_8}}{u} d\rho^2 \right) + \frac{h_8^2}{\Delta^2} ds_{S^3/\mathbf{Z}_k}^2, \\
G_4 =& - d\left( \frac{uu'}{2h_4} + 2\rho h_8 \right) \wedge \text{vol}_{\text{AdS}_3} + 2h_8 d\left( -\rho + \frac{uu'}{4h_4 h_8 + u'^2} \right) \wedge \text{vol}_{S^3/\mathbf{Z}_k} \\
& - h_4' \text{vol}_{CY_2}, \\
\Delta =& \frac{h_8^{1/2}(4h_4 h_8 + u'^2)^{1/3}}{2^{2/3} h_4^{1/6} u^{1/3}},
\end{aligned}
\tag{17}
$$

where $k = h_8$. The quotiented three-sphere is written as a Hopf fibration over S$^2$,

$$ds_{S^3/\mathbf{Z}_k}^2 = \frac{1}{4}\left[ \left( \frac{dz}{k} + \omega \right)^2 + ds_{S^2}^2 \right] \qquad \text{with} \qquad d\omega = \text{vol}_{S^2}. \tag{18}$$

In these solutions, the symmetries SL(2, **R**)$\times$ SL(2, **R**) and SU(2) are geometrically realised by the AdS$_3$ and the quotiented three-sphere, respectively.

In the new class of solutions given by (17), the number of Type IIA D6-branes becomes the orbifold parameter in S$^3/$**Z**$_k$, $k = h_8$. This is associated with the KK-monopole charge. The D2- and D4-branes of the Type IIA solution become M2- and M5-branes, respectively. Their presence is captured by integrating the Page flux $\hat{G}_7 = G_7 - G_4 \wedge C_3$ and a nontrivial flux of $G_4$ through the CY$_2$. In turn, the NS5 branes become M5'-branes. The uplifted brane set-up is the one depicted in Table 2. Recently, it was shown that the solutions emerge in the near horizon limit of this intersecting brane system [36]. The KK-monopoles (wrapped on the CY$_2$) and the M2 branes are stretched between parallel M5'-branes, with extra M5-branes providing for flavour groups.

**Table 2.** $\frac{1}{8}$-BPS brane intersection underlying the AdS$_3 \times$ S$^3/$**Z**$_k$ solutions in M-theory. The 2D CFT lives in the $(x^0, x^1)$ directions, $(x^2, \ldots, x^5)$ span the CY$_2$, $x^6$ is the "field theory" direction and $(x^7, x^8, x^9)$ are the transverse directions on which the SO(3)$_R$ symmetry is realised. Finally, $x^{10}$ is the extra eleventh direction, which spans the S$^1/$**Z**$_k \subset$ S$^3/$**Z**$_k$ and plays the role of the Taub-NUT direction of the KK-monopoles.

|       | 0 | 1 | 2 | 3 | 4 | 5 | 6 | 7 | 8 | 9 | 10 |
|-------|---|---|---|---|---|---|---|---|---|---|----|
| M2    | x | x |   |   |   |   | x |   |   |   |    |
| M5    | x | x |   |   |   |   |   | x | x | x | x  |
| KK    | x | x | x | x | x | x | x |   |   |   | z  |
| M5′   | x | x | x | x | x | x |   |   |   |   |    |

In [34], it was argued that this brane intersection describes the M$_A$-strings introduced in [55,56], supplemented with extra M5-branes. The corresponding dual quivers are the ones depicted in Figure 3, with upper row nodes associated with M2-branes and lower row nodes associated with KK-monopoles. The M5-branes provide extra flavour groups that render the quivers non-anomalous (and the supergravity equations of motion satisfied).

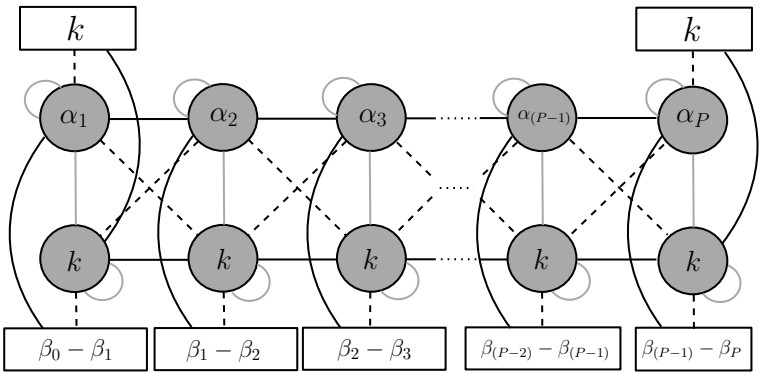

**Figure 3.** Generic quiver field theory in which the IR is dual to the AdS$_3 \times$ S$^3/$**Z**$_k \times$ CY$_2 \times$ I M-theory solutions defined by the functions in (6) and (7).

Thus, the new solutions in M-theory provide for explicit AdS$_3$ geometries where M$_A$-strings can be studied holographically. In particular, the matching between the field theory and holographic computations of the central charge follows directly upon uplift from ten dimensions. The holographic central charge given by Equation (15) becomes, in the massless case,

$$c_{hol} = \frac{3}{\pi} h_8 \int_0^{2\pi(P+1)} \mathrm{d}\rho\, h_4 = 6h_8 \sum_{k=1}^{P} \alpha_k = 6h_8 \sum_{k=1}^{P} Q_{\mathrm{M2}}^{(k)} = 6k Q_{\mathrm{M2}} = 6Q_{\mathrm{M}_A}, \quad (19)$$

where $Q_{\mathrm{M}_A}$ stands for the total number of M$_A$-strings in the configuration, taking into account the orbifolding by **Z**$_k$. This result stresses out that the degrees of freedom of the strongly coupled conformal field theory that originate from the M$_A$-strings.

Furthermore, in [34], a second class of AdS$_3$ solutions to M-theory of the form AdS$_3/$**Z**$_k \times$ S$^3 \times$ CY$_2 \times$ I was obtained through a double analytic continuation from the previous solutions. In the background given in (17), the AdS$_3$ and S$^3$ factors can be swapped as follows:

$$\mathrm{AdS}_3 \to -\mathrm{S}^3, \qquad \mathrm{S}^3 \to -\mathrm{AdS}_3, \quad (20)$$

to produce the new class of solutions, together with

$$u \to -iu, \qquad h_4 \to ih_4, \qquad h_8 \to ih_8, \qquad \rho \to i\rho. \quad (21)$$

These solutions read

$$
ds_{11}^2 = \frac{h_8^2}{\Delta^2} ds_{\mathrm{AdS}_3/\mathbf{Z}_k}^2 + \Delta \left( \frac{u}{\sqrt{h_4 h_8}} ds_{\mathrm{S}^3}^2 + \sqrt{\frac{h_4}{h_8}} ds_{\mathrm{CY}_2}^2 + \frac{\sqrt{h_4 h_8}}{u} d\rho^2 \right)
$$

$$
G_4 = -d\left( -\frac{uu'}{2h_4} + 2\rho h_8 \right) \wedge \mathrm{vol}_{\mathrm{S}^3} - 2h_8 d\left( \rho + \frac{uu'}{4h_4 h_8 - u'^2} \right) \wedge \mathrm{vol}_{\mathrm{AdS}_3/\mathbf{Z}_k} \tag{22}
$$

$$
- h_4' \mathrm{vol}_{\mathrm{CY}_2},
$$

$$
\Delta = \frac{h_8^{1/2}(4h_4 h_8 - u'^2)^{1/3}}{2^{2/3} h_4^{1/6} u^{1/3}},
$$

where $k = h_8$ and the quotiented $\mathrm{AdS}_3$ subspace is written as a Hopf fibration over $\mathrm{AdS}_2$:

$$
ds_{\mathrm{AdS}_3/\mathbf{Z}_k}^2 = \frac{1}{4}\left[ \left( \frac{dz}{k} + \eta \right)^2 + ds_{\mathrm{AdS}_2}^2 \right] \qquad \text{with} \qquad d\eta = \mathrm{vol}_{\mathrm{AdS}_2},
$$

$$
ds_{\mathrm{AdS}_2}^2 = -dt^2 \cosh^2 r + dr^2, \qquad \eta = -\sinh r\, dt. \tag{23}
$$

Notice that the KK-monopoles become M0-branes, with the Taub-NUT direction of the KK-monopoles turned into the direction of propagation of the M0-branes, or waves. These solutions are associated with the M0-M2-M5-M5' brane intersections depicted in Table 3. They preserve the same number of supersymmetries as the original $\mathrm{AdS}_3 \times \mathrm{S}^3/\mathbf{Z}_k \times \mathrm{CY}_2 \times$ I solutions.

**Table 3.** $\frac{1}{8}$-BPS brane intersection underlying the $\mathrm{AdS}_3/\mathbf{Z}_k \times \mathrm{S}^3$ solutions in M-theory. $x^1$ is the direction of propagation of the M0-branes, $(x^2, \ldots, x^5)$ span the $\mathrm{CY}_2$, $x^6$ is the direction along the $\rho$-interval and $(x^7, x^8, x^9, x^{10})$ are the transverse directions on which the SO(4) R-symmetry is realised.

|      | 0 | 1 | 2 | 3 | 4 | 5 | 6 | 7 | 8 | 9 | 10 |
|------|---|---|---|---|---|---|---|---|---|---|----|
| M0   | x | x |   |   |   |   |   |   |   |   |    |
| M2   | x | x |   |   |   |   | x |   |   |   |    |
| M5   | x | x |   |   |   |   |   | x | x | x | x  |
| M5'  | x | x | x | x | x | x |   |   |   |   |    |

## 4. AdS$_2$/CFT$_1$ in Massive IIA

A new class of $\mathrm{AdS}_2 \times \mathrm{S}^3 \times \mathrm{CY}_2 \times$ I solutions to massless Type IIA supergravity can be obtained from (22) upon reduction along the Hopf fibre of the $\mathrm{AdS}_3/\mathbf{Z}_k$ subspace given by (23). These backgrounds are associated with D0-F1-D4-D4' brane intersections that preserve $\mathcal{N} = 4$ supersymmetries in one dimension. These solutions can also be obtained through a double analytical continuation from the solutions reviewed in Section 2. In fact, this allows one to extend them to the massive case. The corresponding brane set-up is depicted in Table 4. In this manner, we find an $\mathrm{AdS}_2 \times \mathrm{S}^3 \times \mathrm{CY}_2 \times$ I class of solutions to massive Type IIA supergravity with the NS-NS sector:

$$
ds^2 = \frac{u}{\sqrt{h_4 h_8}} \left( \frac{h_4 h_8}{4h_4 h_8 - u'^2} ds_{\mathrm{AdS}_2}^2 + ds_{\mathrm{S}^3}^2 \right) + \sqrt{\frac{h_4}{h_8}} ds_{\mathrm{CY}_2}^2 + \frac{\sqrt{h_4 h_8}}{u} d\rho^2,
$$

$$
e^{-2\Phi} = \frac{h_8^{3/2}(4h_4 h_8 - u'^2)}{4h_4^{1/2} u}, \qquad B_2 = -\frac{1}{2}\left( \rho + \frac{uu'}{4h_4 h_8 - u'^2} \right) \mathrm{vol}_{\mathrm{AdS}_2}. \tag{24}
$$

The background is further supported by the RR fluxes:

$$F_0 = h'_8, \quad F_2 = -\frac{1}{2}\left(h_8 + \frac{h'_8 u' u}{4h_8 h_4 - u'^2}\right)\mathrm{vol}_{\mathrm{AdS}_2},$$

$$F_4 = \left(-\mathrm{d}\left(\frac{u' u}{2h_4}\right) + 2h_8\mathrm{d}\rho\right) \wedge \mathrm{vol}_{\mathrm{S}^3} - h'_4\mathrm{vol}_{\mathrm{CY}_2}. \tag{25}$$

The warping functions $h_8$, $h_4$ and $u$ depend on $\rho$, as in the seed solutions. Note that, in this case, $(4h_8 h_4 - u'^2) > 0$ in order to guarantee a real dilaton and a metric with the correct signature. Supersymmetry and the Bianchi identities of the fluxes (away from localised sources) are imposed by Equation (3), which make $h_8$, $h_4$ and $u$ linear functions of $\rho$.

We quote the Page fluxes, $\hat{F} = e^{-B_2} \wedge F$, as follows:

$$\hat{F}_0 = h'_8, \quad\quad\quad \hat{F}_2 = -\frac{1}{2}\left(h_8 - h'_8(\rho - 2\pi k)\right)\mathrm{vol}_{\mathrm{AdS}_2},$$

$$\hat{F}_4 = \left(2h_8\mathrm{d}\rho - \mathrm{d}\left(\frac{u' u}{2h_4}\right)\right) \wedge \mathrm{vol}_{\mathrm{S}^3} - h'_4\mathrm{vol}_{\mathrm{CY}_2}, \tag{26}$$

where we included large gauge transformations of $B_2$ of parameter $k$, $B_2 \to B_2 + \pi k\mathrm{vol}_{\mathrm{AdS}_2}$ (see [40]).

**Table 4.** Brane set-up associated to the solutions (24) and (25). $x^0$ is the time direction of the ten dimensional spacetime; $x^1, \ldots, x^4$ are the coordinates spanning $\mathrm{CY}_2$; $x^5$ is the direction where the F1-strings are stretched; and $x^6, x^7, x^8$, and $x^9$ are the coordinates where the SO(4) R-symmetry is realised.

|      | $x^0$ | $x^1$ | $x^2$ | $x^3$ | $x^4$ | $x^5$ | $x^6$ | $x^7$ | $x^8$ | $x^9$ |
|------|-------|-------|-------|-------|-------|-------|-------|-------|-------|-------|
| D0   | x     |       |       |       |       |       |       |       |       |       |
| D4   | x     | x     | x     | x     | x     |       |       |       |       |       |
| D4′  | x     |       |       |       |       |       | x     | x     | x     | x     |
| D8   | x     | x     | x     | x     | x     |       | x     | x     | x     | x     |
| F1   | x     |       |       |       |       | x     |       |       |       |       |

An infinite family of backgrounds with $u = $ constant and $h_4$ and $h_8$ being piecewise continuous as in (6) and (7) were analysed in [40], and together with their dual SCQM description [5]. We summarise this description in the next subsection.

### 4.1. Dual Superconformal Quantum Mechanics

The superconformal quantum mechanics dual to the previous class of solutions was studied in [40]. The proposal therein is that a 1D $\mathcal{N} = 4$ quantum mechanics lives on the D0-D4-D4′-D8-F1 brane set-up depicted in Table 4 that describes the interactions between D0 and D4 brane instantons and F1 Wilson lines in the 5D gauge theory living in the intersection of D4′ and D8 branes. This is a generalisation of the ADHM quantum mechanics described in [71] and of the quiver proposals discussed in [72,73].

In order to describe the quantum mechanics, the D0-D4-D4′-D8-F1 brane system was split into two subsystems, D4-D4′-F1 and D0-D8-F1, that were first studied independently. The first subsystem was interpreted as describing BPS F1 Wilson lines introduced in the 5D theory living on the D4′-branes by D4-branes [74]. Similarly, the D0-D8-F1 subsystem was interpreted as describing F1 Wilson lines introduced in the worldvolume of the D8-branes by D0-branes [75]. Indeed, the branes in the D4-D4′-F1 and D0-D8-F1 subsystems are displayed exactly as in the D3-D5-F1 brane configuration that describes Wilson lines in antisymmetric representations in 4D $\mathcal{N} = 4$ SYM, studied in [76,77].

For the solutions defined by (6) and (7), the quantised charges associated with the Page fluxes given by (26) in the different $2\pi k \leq \rho \leq 2\pi(k+1)$ intervals are

$$Q_{\mathrm{D4}}^{e(k)} = \alpha_k = \sum_{j=0}^{k-1} \beta_j, \qquad Q_{\mathrm{D0}}^{e(k)} = \mu_k = \sum_{j=0}^{k-1} \nu_j$$

$$Q_{\mathrm{D4}'}^{m(k)} = \beta_k, \qquad Q_{\mathrm{D8}}^{m(k)} = \nu_k, \qquad Q_{\mathrm{F1}}^{e(k)} = 1, \tag{27}$$

where we use electric and magnetic charges, as explained in [40]. The Hanany–Witten brane set-up associated to these brane charges is the one depicted in Figure 4. In order to understand the quantum mechanics associated with this brane configuration, it is useful to go to Type IIB and S-dualise. Then, after performing Hanany–Witten moves, one can go back to Type IIA, where one can interpret the resulting brane set-up (depicted in Figure 5) as describing $\mathrm{U}(\alpha_k)$ and $\mathrm{U}(\mu_k)$ Wilson lines in the completely antisymmetric representations $(\beta_0, \beta_1, \ldots, \beta_{k-1})$ of $\mathrm{U}(\alpha_k)$ and $(\nu_0, \nu_1, \ldots, \nu_{k-1})$ of $\mathrm{U}(\mu_k)$, respectively. Given that the Wilson lines are in the completely antisymmetric representations, the D4-D4'-F1 and D0-D8-F1 subsystems describe baryon vertices [78].

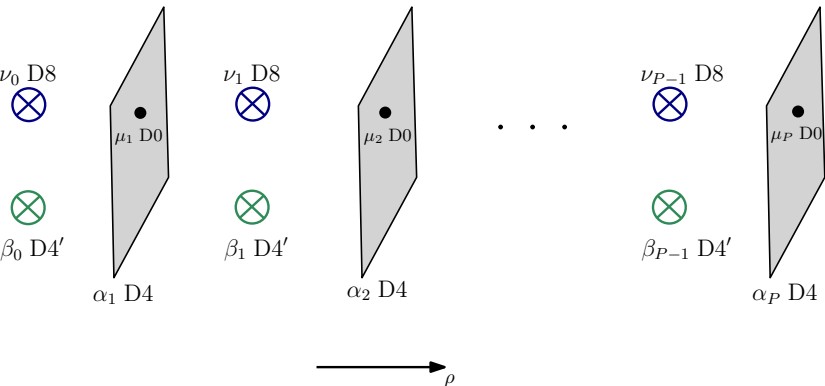

**Figure 4.** Hanany–Witten brane set-up associated with the brane charges of the solutions.

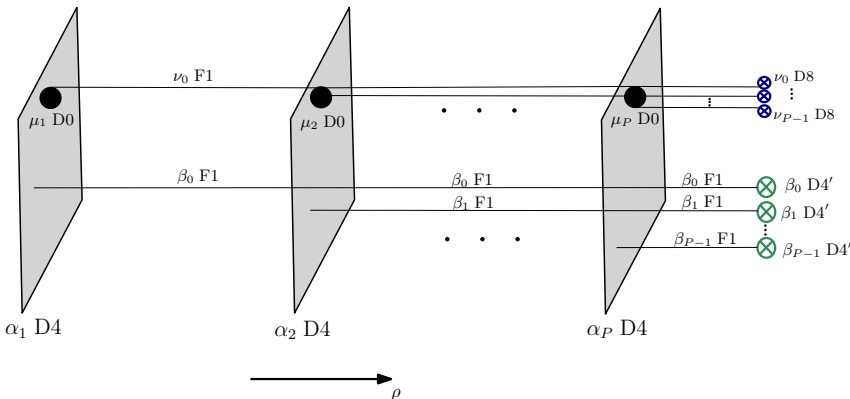

**Figure 5.** Hanany–Witten brane set-up equivalent to the brane configuration in Table 4.

This is consistent with an interpretation of the $\mathrm{AdS}_2$ solutions as describing backreacted baryon vertices within the 5D $\mathcal{N} = 1$ QFT living in D4'-D8 branes. In this interpretation, the SCQM arises in the very low energy limit of a system of D4'-D8 branes, dual to a 5D $\mathcal{N} = 1$ gauge theory, in which one-dimensional defects are introduced. The one-dimensional defects consist of D4-brane baryon vertices, connected to the D4' with F1-strings, and D0-brane baryon vertices, connected to the D8 with F1-strings. In the IR, the gauge symmetry on both the D4' and D8 branes becomes global, turning them from colour to flavour branes. In turn, the D4 and D0 defect branes become the new colour branes of the backreacted geometry. This is in agreement with the defect interpretation found for

this class of solutions in [36], where the AdS$_2$ geometries were shown to asymptote locally to the AdS$_6$ background of Brandhuber–Oz [79].

The previous SCQMs can be given a quiver-like description, which can be used to compute their central charge. From the brane set-up depicted in Figure 4, one can construct the quiver shown in Figure 6, where the gauge groups are associated with the colour D0- and D4-branes and the flavour groups are associated with the D4′- and D8-branes. The quantised charges are the ones computed in Equation (27). The dynamics of the quiver is described in terms of (4, 4) vector multiplets, (4, 4) hypermultiplets in the adjoint representations and (4, 4) hypermultiplets in the bifundamental representations. The connection between colour and flavour branes is through twisted (4, 4) bifundamental hypermultiplets (bent black lines) and (0, 2) bifundamental Fermi multiplets (dashed lines). This follows directly from the analysis of Appendix B in [40].

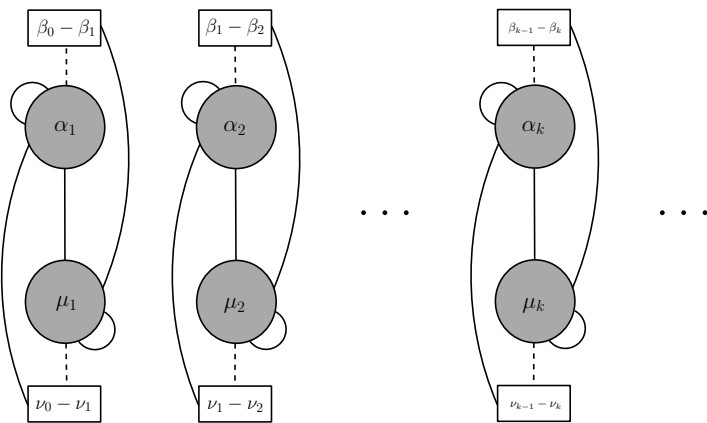

**Figure 6.** One-dimensional quiver field theory in which the IR limit is dual to the AdS$_2$ backgrounds reviewed in Section 4.

As before, a check of the validity of the proposed quivers is given by matching the field theory and holographic central charges. In the case at hand, however, we are dealing with a one-dimensional field theory, for which the definition of central charge is subtle. We interpret the central charge as counting the ground states of the conformal quantum mechanics. In [40], the same expression used in Section 2.1 for the computation of the central charge of the 2D dual CFT was proposed to be valid for the 1D quiver mechanics. For these quivers, $n_{hyp}$ and $n_{vec}$ are, respectively, the numbers of $\mathcal{N} = 4$ (untwisted) hypermultiplets and vector multiplets. Using this expression, perfect agreement was found (in the large number of nodes with large ranks limit) between the quantum mechanics central charge and the holographic central charge, which in this case is obtained through the following integration:

$$c_{\text{hol,1d}} = \frac{3V_{int}}{4\pi G_N} = \frac{3}{4\pi} \frac{\text{Vol}_{CY_2}}{(2\pi)^4} \int_0^{2\pi(P+1)} \left(4h_4 h_8 - u'^2\right) d\rho. \tag{28}$$

This is a striking result, since the superconformal quantum mechanics dual to our class of solutions does not seem to have a direct relation to 2D CFTs. Compared with the results in the literature for the dimension of the Higgs branch of $\mathcal{N} = 4$ quantum mechanics with gauge groups $\prod_v U(N_v)$ connected by bifundamentals [80], one sees that the expression $c_{cft} = 6(n_{hyp} - n_{vec})$ may be interpreted as an extension of the formulas therein to more general $\mathcal{N} = 4$ quivers including flavours. This is an interesting relation that deserves further investigation.

## 5. AdS$_2$ Solutions in Type IIB

In this section, we review two classes of AdS$_2$ solutions to Type IIB supergravity with 4 Poincaré supersymmetries. This is based on the works [38,43]. These solutions are obtained

from the backgrounds reviewed in Sections 2 and 4, respectively, upon T-duality. Moreover, both solutions are related to each other through a double analytical continuation.

The two classes of solutions consist of $AdS_2 \times S^2 \times CY_2 \times S^1$ geometries foliated over an interval. Despite this, they have substantially different dual field theory descriptions that are inherited from their respective T-dual origins. Although we do not review these results in this paper, it was shown in [43] that both types of solutions fit locally in the general class of $AdS_2 \times S^2 \times CY_2 \times \Sigma_2$ solutions of Type IIB supergravity found in [8,9], when there are no D3- and D7-branes present. In the presence of these branes, our solutions extend the previous classifications. In order to show this explicitly, it is necessary to perform a subtle zooming-in that was explained in [43].

### 5.1. Type A

Consider the class given by (1) and (2), where the $AdS_3$ subspace is written as a Hopf fibration over $AdS_2$, as shown by expression (23) for $k = 1$ and $z = \psi$. By applying T-duality on the fibre direction $\psi$, new $AdS_2$ solutions preserving $\mathcal{N} = 4$ supersymmetry are obtained. These backgrounds have the structure of an $AdS_2 \times S^2 \times CY_2 \times S^1$ geometry warped over an interval. The NS-NS sector reads

$$ds^2 = \frac{u}{\sqrt{h_4 h_8}} \left( \frac{1}{4} ds^2_{AdS_2} + \frac{h_4 h_8}{4 h_4 h_8 + u'^2} ds^2_{S^2} \right) + \sqrt{\frac{h_4}{h_8}} ds^2_{CY_2} + \frac{\sqrt{h_4 h_8}}{u} (d\rho^2 + d\psi^2),$$

$$e^{-2\Phi} = \frac{h_8}{4 h_4} \left( 4 h_4 h_8 + u'^2 \right),$$

$$H_3 = \frac{1}{2} d \left( -\rho + \frac{u u'}{4 h_4 h_8 + u'^2} \right) \wedge \text{vol}_{S^2} + \frac{1}{2} \text{vol}_{AdS_2} \wedge d\psi,$$

(29)

where the $h_4$, $h_8$ and $u$ functions are inherited from the backgrounds (1) and (2) and thus have support on $\rho$.

The 10-dimensional RR fluxes are given by

$$F_1 = h_8' d\psi,$$

$$F_3 = -\frac{1}{2} \left( h_8 - \frac{h_8' u' u}{4 h_8 h_4 + u'^2} \right) \text{vol}_{S^2} \wedge d\psi + \frac{1}{4} \left( d \left( \frac{u' u}{2 h_4} \right) + 2 h_8 d\rho \right) \wedge \text{vol}_{AdS_2},$$

$$F_5 = -(1 + \star) h_4' \text{vol}_{CY_2} \wedge d\psi$$

$$= -h_4' \text{vol}_{CY_2} \wedge d\psi + \frac{h_4' h_8 u^2}{4 h_4 (4 h_4 h_8 + u'^2)} \text{vol}_{AdS_2} \wedge \text{vol}_{S^2} \wedge d\rho.$$

(30)

The Type IIB equations of motion are satisfied by imposing the BPS equations and Bianchi identities given by (3). In turn, the Page forms $\hat{F} = e^{-B_2} \wedge F$ are given by

$$\hat{F}_1 = h_8' d\psi,$$

$$\hat{F}_3 = \frac{1}{2} \left( h_8' (\rho - 2\pi k) - h_8 \right) \text{vol}_{S^2} \wedge d\psi + \frac{1}{4} \left( \frac{u' \left( h_4 u' - u h_4' \right)}{2 h_4^2} + 2 h_8 \right) \text{vol}_{AdS_2} \wedge d\rho,$$

$$\hat{F}_5 = \frac{1}{16} \left( \frac{(u - (\rho - 2\pi k) u')(u h_4' - h_4 u')}{h_4^2} + 4(\rho - 2\pi k) h_8 \right) \text{vol}_{AdS_2} \wedge \text{vol}_{S^2} \wedge d\rho$$

$$- h_4' \text{vol}_{CY_2} \wedge d\psi.$$

(31)

The analysis of these fluxes suggests that the brane content that underlies this class of solutions is the one depicted in Table 5. Here, the D1- and D3-branes play the role of colour branes and the D7- and D3-branes play the role of flavour branes. As for the $AdS_3$ solutions reviewed in Section 2, an infinite family of $AdS_2$ backgrounds can be defined by the piecewise linear functions $h_4$ and $h_8$ given by Equations (6) and (7). We turn next to the description of the superconformal quantum mechanics dual to this class of solutions.

**Table 5.** Brane set-up underlying the geometry in (29) and (30). $x^0$ is the time direction, $(x^1, \ldots, x^4)$ span the $CY_2$, $x^5$ is the direction associated with the $\rho$ coordinate, $(x^6, x^7, x^8)$ are the transverse directions realising the SO(3) R-symmetry and $x^9$ is the $\psi$ direction.

|      | 0 | 1 | 2 | 3 | 4 | 5 | 6 | 7 | 8 | 9 |
|------|---|---|---|---|---|---|---|---|---|---|
| D1   | x |   |   |   |   | x |   |   |   |   |
| D3   | x |   |   |   |   |   | x | x | x |   |
| D5   | x | x | x | x | x | x |   |   |   |   |
| D7   | x | x | x | x | x |   | x | x | x |   |
| NS5  | x | x | x | x | x |   |   |   |   | x |
| F1   | x |   |   |   |   |   |   |   |   | x |

Dual Superconformal Quantum Mechanics

The $\mathcal{N} = 4$ superconformal quantum mechanics dual to the previous class of solutions was studied in [38]. At the geometrical level, these solutions are related to the $AdS_3$ solutions reviewed in Section 2 through T-duality. At the level of the dual CFTs, the superconformal quantum mechanics dual to the $AdS_2$ solutions should then arise from the 2D CFTs dual to the $AdS_3$ backgrounds upon dimensional reduction.

More concretely, in the coordinates used to obtain the $AdS_2$ geometry, defined in Equation (23), the boundaries of $AdS_3$ are two null cylinders [63]. For this reason, the 2D CFT that lives at these boundaries is effectively discrete light-cone quantised (DLCQ) because just one of the $SL(2, \mathbf{R}) \times SL(2, \mathbf{R})$ sectors of global $AdS_3$ survives the compactification. T-dualisation in these coordinates is then equivalent to starting with a given $\mathcal{N} = (0, 4)$ 2D CFT, such as those described in Section 2.1, and being discrete light-cone quantised, keeping $\mathcal{N} = 4$ as the SUSY right sector. This provides an explicit realisation of the constructions in [62–65]. Field theoretically, we start with the Lagrangian describing the 2D CFT dual to $AdS_3$ and dimensionally reduce it to a matrix model where only the time dependence and the zero modes in the T-dual direction are kept.

The concrete proposal in [38] is that the dynamics of the UV quantum mechanics is decribed by the dimensional reduction along the space-direction of the 2D QFTs discussed in Section 2.1. The quiver field theories are then the same ones depicted in Figure 2 but now associated with D1 and D5 colour branes and D3 and D7 flavour branes. In turn, the matter fields are $\mathcal{N} = 4$ multiplets, realised as dimensionally reduced 2D $(0, 4)$ multiplets. Note that these quivers inherit the anomaly cancellation condition of the 2D quivers, even if there is no anomaly cancellation condition in 1D.

As in previous AdS/CFT pairs, one can check the agreement between the field theory and holographic central charges to test the proposed duality. In this case, the usage of expression (14) for computing the quantum mechanics central charge is fully justified, since it arises from a 2D CFT upon compactification. As expected, perfect agreement is found with the holographic central charge, which is computed from the same expression (15) given its invariance under T-duality.

### 5.2. Type B

In this subsection, we review the $AdS_2$ solutions to Type IIB supergravity that arise from the backgrounds defined in (24) and (25) upon T-duality along the Hopf-fibre of the three sphere. The resulting class of solutions have an NS-NS sector

$$ds^2 = \frac{u\sqrt{h_4 h_8}}{4h_4 h_8 - u'^2} ds^2_{AdS_2} + \frac{u}{4\sqrt{h_4 h_8}} ds^2_{S^2} + \sqrt{\frac{h_4}{h_8}} ds^2_{CY_2} + \frac{\sqrt{h_4 h_8}}{u}(d\psi^2 + d\rho^2),$$

$$e^{-2\Phi} = \frac{h_8}{4h_4}(4h_4 h_8 - u'^2),$$

$$H_3 = -\frac{1}{2}d\left(\rho + \frac{uu'}{4h_4 h_8 - u'^2}\right) \wedge \text{vol}_{AdS_2} + \frac{1}{2}\text{vol}_{S^2} \wedge d\psi,$$

(32)

and RR fluxes,

$$
\begin{aligned}
F_1 &= h_8' \mathrm{d}\psi, \\
F_3 &= -\frac{1}{2}\left(h_8 + \frac{h_8' u' u}{4 h_8 h_4 - u'^2}\right)\mathrm{vol}_{\mathrm{AdS}_2}\wedge\mathrm{d}\psi + \frac{1}{4}\left(-\mathrm{d}\left(\frac{u' u}{2h_4}\right) + 2h_8\mathrm{d}\rho\right)\wedge\mathrm{vol}_{\mathrm{S}^2}, \\
F_5 &= -(1 + \star_{10})h_4'\mathrm{vol}_{\mathrm{CY}_2}\wedge\mathrm{d}\psi, \\
&= -h_4'\mathrm{vol}_{\mathrm{CY}_2}\wedge\mathrm{d}\psi - \frac{h_8 h_4' u^2}{4h_4(4h_8 h_4 - u'^2)}\mathrm{vol}_{\mathrm{AdS}_2}\wedge\mathrm{vol}_{\mathrm{S}^2}\wedge\mathrm{d}\rho.
\end{aligned}
\tag{33}
$$

Here, $\psi \in [0, 2\pi]$ is the T-dual coordinate. Note that $4h_4 h_8 - u'^2 \geq 0$ must be imposed to have well-defined supergravity fields. Supersymmetry holds whenever $u'' = 0$. In the same way, the Bianchi identities of the fluxes impose $h_8'' = 0$ and $h_4'' = 0$, away from localised sources. The $\rho$ coordinate describes an interval that we take to be bounded between 0 and $2\pi(P+1)$, as in the previous sections.

The Page fluxes read

$$
\begin{aligned}
\hat{F}_1 &= h_8' \,\mathrm{d}\psi, \\
\hat{F}_3 &= \frac{1}{2}\left(h_8'(\rho - 2\pi k) - h_8\right)\mathrm{vol}_{\mathrm{AdS}_2}\wedge\mathrm{d}\psi + \frac{1}{4}\left(2h_8 + \frac{u'(uh_4' - h_4 u')}{2h_4^2}\right)\mathrm{vol}_{\mathrm{S}^2}\wedge\mathrm{d}\rho, \\
\hat{F}_5 &= \frac{1}{4}\left(h_8(\rho - 2\pi k) - \frac{(u - (\rho - 2\pi k)u')(uh_4' - h_4 u')}{4h_4^2}\right)\mathrm{vol}_{\mathrm{AdS}_2}\wedge\mathrm{vol}_{\mathrm{S}^2}\wedge\mathrm{d}\rho \\
&\quad - h_4'\mathrm{vol}_{\mathrm{CY}_2}\wedge\mathrm{d}\psi.
\end{aligned}
\tag{34}
$$

The analysis of these fluxes yields the brane set-up summarised in Table 6 as underlying this class of solutions. Here, the D1- and D5-branes play the role of colour branes and the D3- and D7-branes of flavour branes. Both the brane set-up and the warped $\mathrm{AdS}_2\times\mathrm{S}^2\times\mathrm{CY}_2\times\mathrm{S}^1$ structure of this class of solutions are shared with those of the solutions reviewed in the previous section. The precise relation between the two types of backgrounds is through the double analytic continuation:

$$
\begin{aligned}
\mathrm{ds}_{\mathrm{AdS}_2}^2 &\to -\mathrm{ds}_{\mathrm{S}^2}^2, & \mathrm{ds}_{\mathrm{S}^2}^2 &\to -\mathrm{ds}_{\mathrm{AdS}_2}^2, & e^{\Phi} &\to ie^{\Phi}, & F_i &\to -iF_i, \\
u &\to -iu, & h_4 &\to ih_4, & h_8 &\to ih_8, & \rho &\to i\rho, & \psi &\to -i\psi.
\end{aligned}
\tag{35}
$$

**Table 6.** Brane set-up underlying the geometry in (32) and (33). $x^0$ is the time direction; $x^1, \ldots, x^4$ are the coordinates spanned by the $\mathrm{CY}_2$; $x^5$ is the direction associated with the $\rho$ coordinate; $(x^6, x^7, x^8)$ are the transverse coordinates realising the SO(3) R-symmetry; and $x^9$ is the $\psi$ direction.

|      | 0 | 1 | 2 | 3 | 4 | 5 | 6 | 7 | 8 | 9 |
|------|---|---|---|---|---|---|---|---|---|---|
| D1   | x |   |   |   |   |   |   |   |   | x |
| D3   | x |   |   |   |   |   | x | x | x |   |
| D5   | x | x | x | x | x |   |   |   |   | x |
| D7   | x | x | x | x | x |   | x | x | x |   |
| NS5  | x | x | x | x | x | x |   |   |   |   |
| F1   | x |   |   |   |   | x |   |   |   |   |

### 5.2.1. Dual Superconformal Quantum Mechanics

The superconformal quantum mechanics dual to this new class of solutions was discussed in [43]. Given that they are obtained via T-duality from the class of $\mathrm{AdS}_2$ solutions reviewed in Section 4.1, they should be dual to the same superconformal quantum mechanics. In this case, the Wilson lines arise from the massive fermionic strings that stretch between D1-branes (and D5-branes) in the $k$th interval and D7-branes (and D3-branes) in all other intervals. In turn, the Wilson lines are in the $(\nu_0, \ldots, \nu_{k-1})$ and $(\beta_0, \ldots, \beta_{k-1})$

completely antisymmetric representation of the U($\mu_k$) and U($\alpha_k$) gauge groups, respectively. As we indicated in Section 4.1, given that the Wilson lines are in the completely antisymmetric representation, the D1-D7-F1 and D5-D3-F1 subsystems describe baryon vertices [78].

This is consistent with an interpretation of our AdS$_2$ solutions as describing backreacted baryon vertices within the 4D $\mathcal{N} = 2$ QFT living in D3-D7 branes. In this interpretation, the SCQM arises in the very low energy limit of a system of D3-D7 branes, dual to a 4D $\mathcal{N} = 2$ QFT, in which one-dimensional defects are introduced. The one-dimensional defects consist of D5-brane baryon vertices, connected to the D3 with F1-strings, and D1-brane baryon vertices, connected to the D7 with F1-strings. In the IR, the gauge symmetry on both the D7 and D3 branes becomes global, turning them from colour to flavour branes. In turn, the D5 and D1 defect branes become the new colour branes of the backreacted geometry. It would be very interesting to explicitly realise this defect interpretation geometrically.

## 6. Discussion

In this review article, we summarised the salient features of the recent developments in [26,34,38,40,43,52–54], in relation to the construction of AdS$_3$/CFT$_2$ and AdS$_2$/CFT$_1$ pairs with four supercharges. For clarity, we summarised the connections between these new classes of solutions in Figure 7.

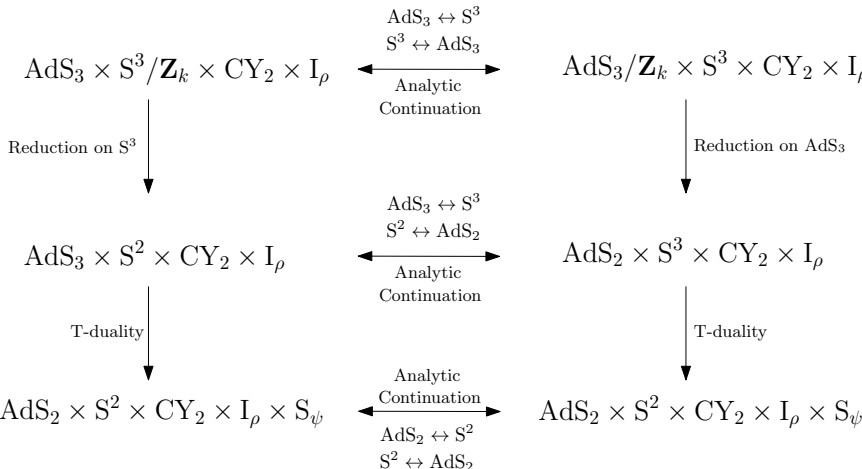

**Figure 7.** Connections between the different classes of solutions reviewed in this paper.

The construction of these new dual pairs extends existing classifications of AdS$_3$ and AdS$_2$ solutions to the case with four supersymmetries. Moreover, the analysis in these references complements the construction of the backgrounds with a comprehensive study of the 2D and 1D CFTs dual to them. The proposed CFTs are described in the UV by means of explicit quivers from which observables such as the central charge can be computed and checked against holographic calculations. The holographic central charge is interpreted as the entanglement entropy of black strings or black holes with AdS$_3$ or AdS$_2$ near horizon geometries. This can then be cross-checked against the field theory computation, within a well controlled setting where the microstate counting programme can be put to work. This line of research remains to be further exploited. See [81–83]. In particular, it would be interesting to apply exact calculational techniques (see [84]) to the new classes of solutions, since this would provide for a deeper understanding of the IR regime of the different theories.

A study that would be interesting to pursue further is the interpretation of the new classes of solutions as surface or line-defect CFTs within higher dimensional conformal field theories. Recent progress in this direction has shown that some subclasses of the AdS$_3$ and AdS$_2$ solutions to massive IIA supergravity (constructed in Sections 2 and 4) asymptote locally to the AdS$_6$ background of Brandhuber–Oz [79]. This means that they can be

interpreted as surface or line-defect CFTs, respectively, within the 5D Sp(N) fixed point theory dual to the Brandhuber–Oz solution [36]. In the AdS$_2$ case, this is in nice agreement with our proposed interpretation of these solutions as backreacted D4-D0 baryon vertices in a system of D4′-D8 branes. In this description, the one-dimensional defects consist of D4-D0 baryon vertices connected to the D4′-D8 branes with (fermionic) F1-strings. In the IR, the gauge symmetry on the D4′-D8 branes becomes global, turning them from colour to flavour branes. In turn, the D4-D0 defect branes become the new colour branes of the backreacted geometry.

In order to complete the previous picture, we should keep in mind that D4-D8 brane set-ups must include O8 orientifold fixed planes in order to flow to a 5D fixed point theory in the UV [85]. It would be interesting to clarify to what extent the behaviour found in [40] at both ends of the space, compatible with the presence of O8 orientifold fixed planes, would provide for a fully consistent global picture. It is expected that, in this set-up, baryon vertices affected by the orbifold projection are removed from the spectrum, corresponding to the fact that US$p$ baryons are unstable against their decay into mesons. A similar interpretation for the D2-D6-NS5 brane surface defects within the D4′-D8 brane intersection, put forward in [36] for the AdS$_3$ case, still remains to be elucidated. In both the AdS$_2$ and AdS$_3$ cases, an explicit realisation of the quiver CFTs as embedded in the 5D quiver CFT associated with the D4′-D8 brane system remains to be found.

This is in contrast with the interpretation of a subclass of the M-theory pairs reviewed in Section 3 as surface defects within the 6D (1, 0) CFT living in M5-branes on ALE singularities, found in [36]. In this case, it has been possible to explicitly realise the 2D quiver CFTs as embedded in the 6D quiver CFT associated with the M5-branes intersecting with KK-monopoles.

We argued that the Type B AdS$_2$ solutions reviewed in Section 5.2 describe backreacted D5-D1 baryon vertices in the 4D $\mathcal{N} = 2$ QFT living in D3-D7 intersections. In this case, there is no holographic analogue, and it would be interesting to see if these solutions asymptote locally to an AdS$_5$ background. This would provide further support to the proposed defect interpretation.

Finally, we stress that the full class of AdS$_3$ solutions discussed in [26,52–54], which constitute the basis for the developments reviewed in this paper, is much broader than the subset of solutions that have been the focus of our CFT analysis. In particular, there is evidence that interesting new CFTs arise when there is dependence on the internal structure of the CY$_2$ manifold. Work is in progress in this direction [69].

**Funding:** This research was partially funded by the Spanish government grant number PGC2018-096894-B-100. AR is supported by CONACyT-Mexico.

**Institutional Review Board Statement:** Not applicable.

**Informed Consent Statement:** Not applicable.

**Acknowledgments:** We thank Niall Macpherson, Carlos Nunez and Stefano Speziali for their collaboration in some of the results reviewed in this article and especially Chris Couzens, Niall Macpherson and Carlos Nunez for careful reading of the manuscript. We are indebted to Prof. Norma Sanchez for inviting us to contribute this review article to the Open Access Special Issue "Women Physicists in Astrophysics, Cosmology and Particle Physics" to be published in *Universe* (ISSN 2218-1997, IF 1.752).

**Conflicts of Interest:** The authors declare no conflicts of interest.

## Notes

1.  By which we mean one and two dimensional.
2.  The $u \neq$ constant case was studied in [37,54].
3.  This is also compactible with a superposition of O2-O6 planes. The string theory interpretation of smeared orientifold fixed planes is however unclear.
4.  See [66–68] for further developments.
5.  A concrete example with $u' \neq 0$ was analysed in [45].

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
