# Peer review of "New Advancements in AdS/CFT in Lower Dimensions"

_universe, doi:10.3390/universe7070250_

Round 1

Reviewer 1 Report

This paper reviews a series of works done by the authors recently, concerning a new class of AdS/CFT examples where explicit string theory construction and the field theory analysis are possible. The range of reviewed works is fairly comprehensive, since massive IIA, M-theory, and also IIB constructions are covered. I recommend highly this work to be published immediately. 

Author Response

We would like to thank the referee for his/her nice comments about our review.

Reviewer 2 Report

The authors present a review of the salient features of recent results obtained by themselves and collaborators on AdS/CFT in low dimensions. The several cases are summarized clearly and connected with each other. Clearly, the article is suitable for this special issue.

Author Response

(The authors gave the same response as above.)

Reviewer 3 Report

This manuscript is a thorough and well written review of AdS/CFT in lower dimensions. This review will be useful for the community. I recommend it for publication.

Author Response

(The authors gave the same response as above.)
